# Mesolimbic confidence signals guide perceptual learning in the absence of external feedback

**Matthias Guggenmos[1,2]\*, Gregor Wilbertz[2], Martin N Hebart[3†], Philipp Sterzer[1,2†]**

[1]Bernstein Center for Computational Neuroscience, Berlin, Germany; [2]Visual Perception Laboratory, Charité Universitätsmedizin, Berlin, Germany; [3]Department of Systems Neuroscience, Universitätsklinikum Hamburg-Eppendorf, Hamburg, Germany

**Abstract** It is well established that learning can occur without external feedback, yet normative reinforcement learning theories have difficulties explaining such instances of learning. Here, we propose that human observers are capable of generating their own feedback signals by monitoring internal decision variables. We investigated this hypothesis in a visual perceptual learning task using fMRI and confidence reports as a measure for this monitoring process. Employing a novel computational model in which learning is guided by confidence-based reinforcement signals, we found that mesolimbic brain areas encoded both anticipation and prediction error of confidence—in remarkable similarity to previous findings for external reward-based feedback. We demonstrate that the model accounts for choice and confidence reports and show that the mesolimbic confidence prediction error modulation derived through the model predicts individual learning success. These results provide a mechanistic neurobiological explanation for learning without external feedback by augmenting reinforcement models with confidence-based feedback.

**\*For correspondence:** matthias.guggenmos@charite.de

[†]These authors contributed equally to this work

**Competing interests:** The authors declare that no competing interests exist.

## Introduction

Learning is an integral part of our everyday life and necessary for survival in a dynamic environment. The behavioral changes arising from learning have quite successfully been described by the reinforcement learning principle (*Sutton and Barto, 1998*), according to which biological agents continuously adapt their behavior based on the consequences of their actions. Thus, reinforcement learning models and most other learning models depend on feedback from the environment. Yet, there are important instances of learning where no such external feedback is provided, challenging the generality of these learning models in shaping our behavior.

A well-studied case of learning is the improvement of performance in perceptually demanding tasks through training or repeated exposure (*Gibson, 1963*). Such *perceptual learning* has repeatedly been demonstrated to occur without feedback (*Herzog and Fahle, 1997*; *Gibson and Gibson, 1955*; *McKee and Westheimer, 1978*; *Karni and Sagi, 1991*) and is therefore ideally suited as a test case to study learning in the absence of external feedback. Previous work has emphasized the role of reinforcement learning in perceptual learning (*Kahnt et al., 2011*; *Law and Gold, 2009*). However, these accounts were based on perceptual learning *with* external feedback and therefore cannot account for instances in which learning occurs *without* external feedback. Here, we pursued the idea that, in the absence of external feedback, learning is guided by internal feedback processes that evaluate current perceptual information in relation to prior knowledge about the sensory world. We reasoned that introspective reports of perceptual confidence could serve as a window into such internal feedback processes. In this scenario, low or high confidence would correspond to a negative

**eLife digest** Much of our behavior is shaped by feedback from the environment. We repeat behaviors that previously led to rewards and avoid those with negative outcomes. At the same time, we can learn in many situations without such feedback. Our ability to perceive sensory stimuli, for example, improves with training even in the absence of external feedback.

Guggenmos et al. hypothesized that this form of perceptual learning may be guided by self-generated feedback that is based on the confidence in our performance. The general idea is that the brain reinforces behaviors associated with states of high confidence, and weakens behaviors that lead to low confidence.

To test this idea, Guggenmos et al. used a technique called functional magnetic resonance imaging to record the brain activity of healthy volunteers as they performed a visual learning task. In this task, the participants had to judge the orientation of barely visible line gratings and then state how confident they were in their decisions.

Feedback signals derived from the participants' confidence reports activated the same brain areas typically engaged for external feedback or reward. Moreover, just as these regions were previously found to signal the difference between actual and expected rewards, so did they signal the difference between actual confidence levels and those expected on the basis of previous confidence levels. This parallel suggests that confidence may take over the role of external feedback in cases where no such feedback is available. Finally, the extent to which an individual exhibited these signals predicted overall learning success.

Future studies could investigate whether these confidence signals are automatically generated, or whether they only emerge when participants are required to report their confidence levels. Another open question is whether such self-generated feedback applies in non-perceptual forms of learning, where learning without feedback has likewise been a long-standing puzzle.

or positive self-evaluation of one's own perceptual performance, respectively. Accordingly, confidence could act as a teaching signal in the same way as external feedback in normative theories of reinforcement learning (*Daniel and Pollmann, 2012*; *Hebart et al., 2014*). Applied to the case of perceptual learning, a confidence-based reinforcement signal could serve to strengthen neural circuitry that gave rise to high-confidence percepts and weaken circuitry that led to low-confidence percepts, thereby enhancing the quality of future percepts.

We tested this idea in a challenging perceptual learning task, in which participants continuously reported their confidence in perceptual choices while undergoing functional magnetic resonance imaging (fMRI). No external feedback was provided; instead, confidence ratings were used as a proxy of internal monitoring processes. To account for perceptual learning in the absence of feedback, we devised a confidence-based associative reinforcement learning model. In the model, confidence prediction errors (*Daniel and Pollmann, 2012*) serve as teaching signals that indicate the mismatch between the current level of confidence and a running average of previous confidence experiences (expected confidence). Based on recent evidence of confidence signals in the mesolimbic dopamine system (*Daniel and Pollmann, 2012*; *Hebart et al., 2014*; *Schwarze et al., 2013*), we hypothesized to find neural correlates of confidence prediction errors in mesolimbic brain areas such as the ventral striatum and the ventral tegmental area. Since confidence prediction errors act as a teaching signal in our model, we hypothesized that the strength of these mesolimbic confidence signals should be linked to individual perceptual learning success.

## Results

Human participants (N=29) learned to detect the orientation of peripheral noise-embedded Gabor patches relative to a horizontal or vertical reference axis while undergoing functional magnetic resonance imaging (fMRI). Overall, the experiment comprised four sessions: (i) an initial behavioral test session to establish participants' baseline contrast thresholds for a performance level of 80.35% correct responses, (ii) an intensive perceptual learning session (training) in the MRI scanner with a continuous threshold determination, and two behavioral post-training test sessions to examine

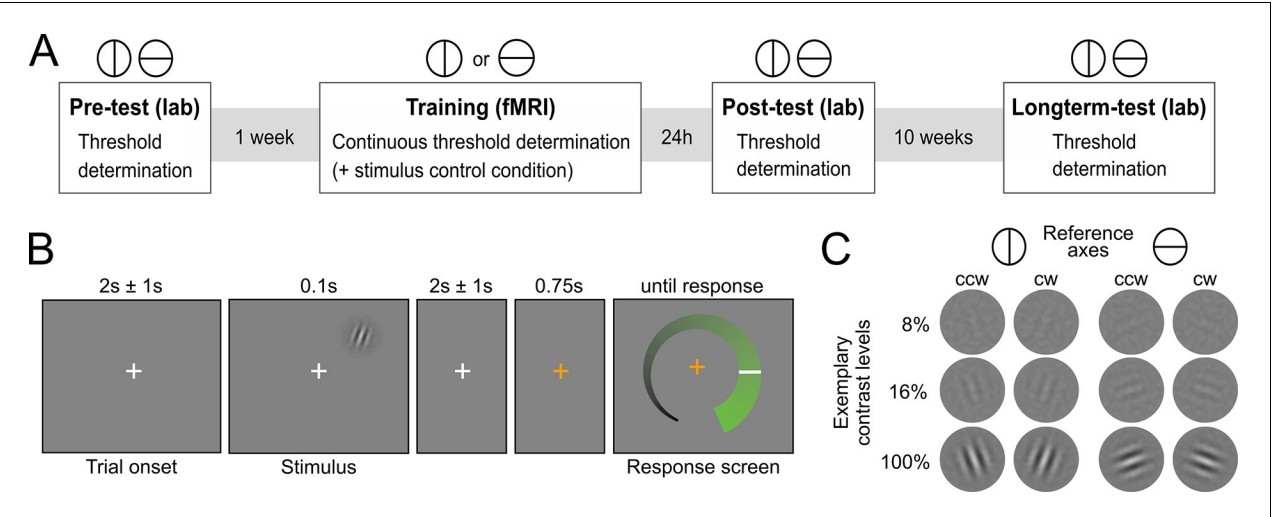

**Figure 1.** Experimental design. (A) Overview over experimental sessions. The experiment consisted of one training session and three test sessions (pre-test, post-test and longterm-test). The test sessions included both reference axes and were used to determine the contrast threshold for a performance of 80.35 percent correct at different stages of the experiment. In the training session, only one reference axis was shown. Here too, a staircase procedure was used to continuously determine the contrast threshold for a performance level of 80.35%. In addition, the training session included a condition with constant contrast as a control for stimulus factors. (B) Procedure of an experimental trial. Participants were presented with Gabor stimuli, which were oriented either clockwise or counterclockwise with respect to a reference axis. In the unspeeded response phase participants indicated their level of confidence about the stimulus orientation on an analogue scale and subsequently made a binary orientation judgment. (C) Examples of the stimuli. Gabor patches were oriented 20° clockwise (cw) or 20° counterclockwise (ccw) relative to either the vertical or the horizontal reference axis. Three exemplary contrast levels are shown, where 8% corresponds to the participant average during training, 16% to the highest obtained thresholds and 100% to full contrast.

(iii) short-term and (iv) long-term stimulus-specific training effects (*Figure 1A*). While the training session was based on one reference axis, all test sessions comprised a contrast threshold determination for both reference axes. The training session additionally included a control condition in interleaved presentation, for which the contrast was kept constant to enable an exploratory multivariate analysis of changes in neural stimulus representation. The Gabor stimuli were flashed briefly in the upper right quadrant and participants had to judge their orientation with respect to the current reference axis (*Figures 1B,C*). Eyetracking ensured that participants maintained fixation throughout the training session (*Figure 2—figure supplement 1*). Importantly, participants did not receive external cognitive or rewarding feedback during the entire experiment. Rather, in addition to their choice, they reported their confidence about the stimulus orientation on a visual analogue scale (for a verification of accurate usage, see *Figure 2—figure supplement 2*). The confidence reports were used to compute the internal feedback in our model on a trial-by-trial basis.

## Stimulus-specific perceptual learning

To establish stimulus-specific perceptual learning, we compared perceptual thresholds in pre- and post-experimental sessions between the trained and untrained reference axis (*Figure 2A*). The contrast thresholds improved for the trained ($t_{28}$ = 6.73, p < 0.001, two-tailed), but not for the untrained reference axis ($t_{28}$ = 0.41, p = 0.68; interaction of training × time: $F_{1,28}$ = 14.2, p < 0.001), demonstrating clear and specific effects of perceptual learning. These stimulus-specific training effects could still be detected 10 weeks later ($F_{1,28}$ = 4.3, p = 0.047), indicating long-term stability and thus demonstrating a key characteristic of perceptual learning (*Karni and Sagi, 1993*). To test whether the effects of learning could already be detected during the training session, we linearly fitted the contrast thresholds across trials in the critical constant performance condition. The analysis showed that contrast threshold consistently decreased across runs (linear slope: −0.006 ± 0.002, $t_{28}$ = −2.38, p = 0.024), from 8.64% ± 0.47 (mean ± SEM) in the first training run to 7.68% ± 0.52 in the last training run.

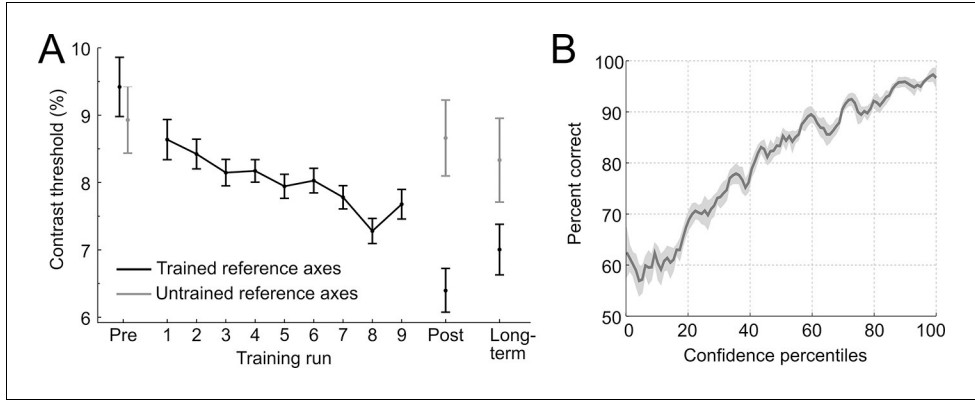

**Figure 2.** Behavioral results. (**A**) Contrast thresholds across the runs of the training session and in the three test-sessions (pre/post/long-term). (**B**) Relationship between confidence ratings and performance during the training session. Percent correct responses were computed by means of a sliding window across sorted confidence values (window size: 5% of all trials).
The following figure supplements are available for figure 2:

**Figure supplement 1.** Eyetracking.
**Figure supplement 2.** Confidence ratings.

To assess whether participants adequately used the continuous confidence rating scale during training, the relationship between reported confidence and performance (proportion correct) was analyzed by means of a sliding window across sorted confidence values. The performance increased monotonically with confidence (main effect of percentile: $F_{22,2442} = 5.76$, $p < 0.001$, one-way ANOVA with repeated measures), approaching chance at low levels of confidence, without showing ceiling effects at high levels of confidence (*Figure 2B*). This pattern indicates a close link between the decisional certainty of the choice and the reported confidence and shows that confidence represents valuable self-generated feedback.

## A confidence-based model of perceptual learning

To account for perceptual learning without external feedback, we devised an associative reinforcement learning model with confidence as internal feedback (*Figure 3*). Learning in this model was guided by the combination of a confidence-based reinforcement signal and Hebbian plasticity, inspired by the previously proposed *three-factor learning rule* (dopaminergic reinforcement signal, pre-synaptic activity, post-synaptic activity) for neural plasticity in the mesolimbic system (*Reynolds et al., 2001*; *Schultz, 2002*). Our model assumes that observers improve perceptual performance by optimizing a filter on incoming sensory evidence. The filter is represented by two components: *signal weights* for clockwise (cw) and counterclockwise (ccw) stimulus orientation ($w_{ccw,ccw}$; $w_{cw,cw}$), connecting orientation energy detectors $E_{ccw/cw}$ to decision units $A_{ccw/cw}$ with same orientations; and *noise weights* ($w_{ccw,cw}$; $w_{cw,ccw}$), connecting detectors $E_{ccw/cw}$ to decision units $A_{cw/ccw}$ with opposing orientations. The clockwise and counterclockwise orientation energy contained in the stimuli is computed by a simple model of primary visual cortex (*Petrov et al., 2005*). The weighted sums of $E_{ccw/cw}$ determine the activities of decision units $A_{cw/ccw}$ which, in a next step, are integrated in a *decision value* $DV = A_{cw} - A_{ccw}$. $DV$ translates into probabilities for clockwise or counterclockwise choices via a softmax action selection rule, and to the model's equivalent of confidence—*decisional certainty*—through its absolute value. Finally, perceptual learning is based on an associative reinforcement learning rule with two separate components: a reinforcement component utilizes a confidence prediction error (CPE; denoted as $\delta$) as internal feedback, representing the mismatch between current confidence and a long-term estimate of expected confidence (via a learning rate $\alpha_c$); and a Hebbian component ensures that the weights are updated in proportion to how strongly orientation detectors and decision units co-activate. In addition, a learning rate $\alpha_w$ accounts for

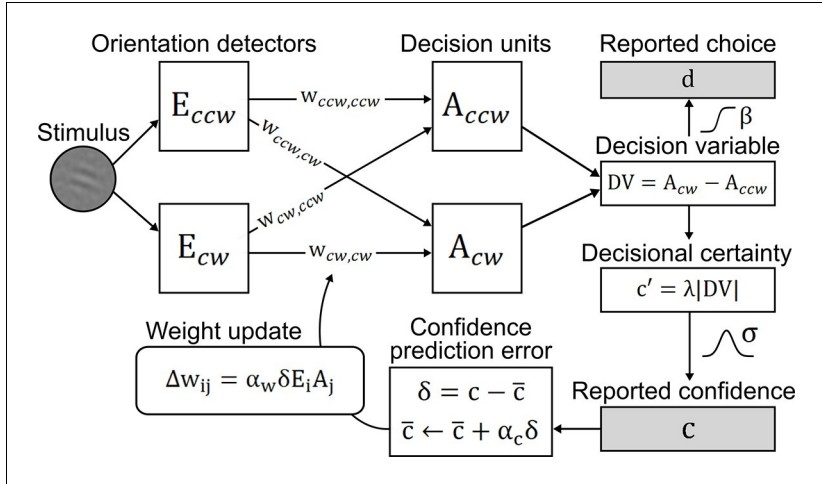

**Figure 3.** Confidence-based model of perceptual learning. Counterclockwise ($E_{cw}$) and clockwise ($E_{ccw}$) orientation energy detectors of a dedicated representational subsystem are connected via *signal weights* (horizontal) and *noise weights* (diagonal) to decision units ($A_{ccw}$, $A_{cw}$). Reported choices (decisions) *d* are probabilistically modeled by a *decision value* $DV = A_{ccw} - A_{cw}$ and the reported confidence *c* is modeled through the absolute value of *x*. Weights are updated through an associative reinforcement learning update rule. The reinforcement component is based on a *confidence prediction error δ*, reflecting the difference between reported confidence and a weighted running average of previous confidence experiences (*expected confidence* $\bar{c}$). The Hebbian component ($E_i \times A_j$) ensures that the update more strongly affects those connections that contribute more to the final choice. Grey-shaded boxes indicate observed variables.

The following figure supplement is available for figure 3:

**Figure supplement 1.** Exemplary time course of model variables and behavioral reports.

---

inter-individual differences in learning speed. *Figure 3—figure supplement 1* provides an exemplary time slice of one participant's behavioral reports and accompanying model variables.

In a first step, we validated the representational subsystem by computing the average energy content for a range of spatial frequencies (one octave above and below the actual frequency of 1.25 cycles/degree) and for a range of orientations (−40° to +40° relative to the reference axes). As expected, the energy content was higher for the spatial frequency and orientations used to generate the Gabor patches relative to other frequencies and orientations (*Figure 4—figure supplement 1*). Further, when orientation energy was computed separately for correct and incorrect responses, the energy for *designated orientations* (i.e., the orientations used to generate the Gabor patches) was *higher for correct than for incorrect responses* ($t_{28} = 8.1$, $p < 0.001$), whereas the energy for *opposite orientations* (i.e., ∓20° if ±20° was presented) was *lower for correct than for incorrect responses* ($t_{28} = -3.6$, $p = 0.001$) (*Figure 4A*). This pattern demonstrated that the varying orientation energy was directly associated with behavior and adds to the validation of the model. This pattern did also hold when the analysis was restricted to trials of the constant contrast condition (designated orientation: $t_{28} = 6.64$, $p < 0.001$; opposite orientation: $t_{28} = -2.38$, $p = 0.024$), thereby showing that the representational subsystem accounted for additional variance due to the random noise field over and above the variance due to changes in stimulus contrast.

We then went on to fit the model parameters to participants' orientation and confidence reports in the training session using maximum likelihood approximation (median ± SE of the median: $\alpha_w = 0.0018 \pm 0.0007$, $\alpha_c = 0.533 \pm 0.077$; see *Supplementary file 1* for other parameters). To assess the model fit, we correlated model-based choice probabilities with participants' actual choices, separately for each participant. This analysis showed that the model accounted well for participants' choices (mean ± SE of individual z-transformed correlation coefficients: $r_{Pearson} = 0.64 \pm 0.03$; one-sample t-test against Fisher z' = 0: $t_{28} = 26.2$, $p < 0.001$). This correspondence is reflected in the fact that participants' and model-based choice probabilities show a nearly identical (sigmoidal) dependency on *DV* (*Figure 4A*; see *Figure 4—figure supplement 2* for single-subject fits). We next

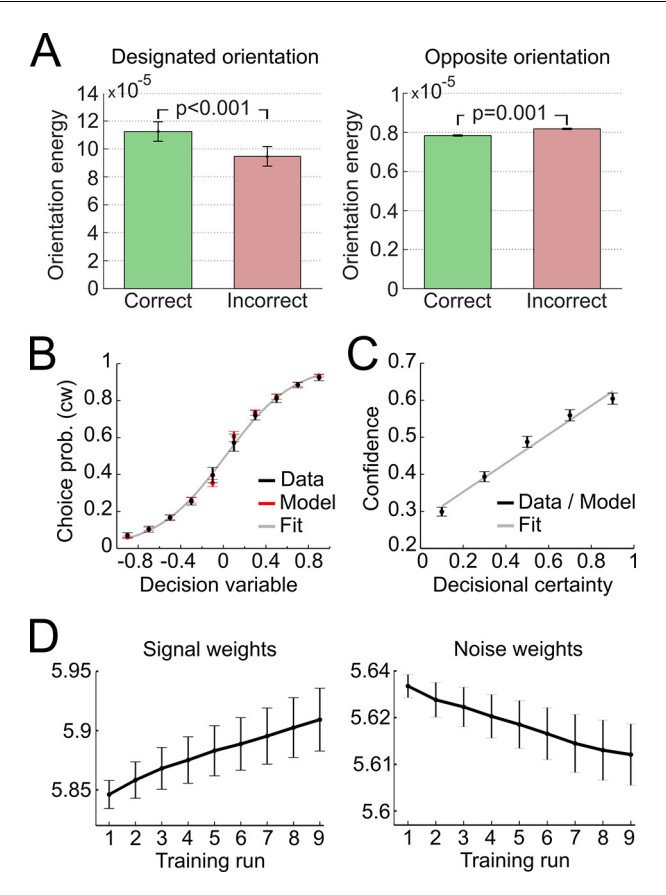

**Figure 4.** Modeling results. (**A**) Orientation energy computed by the model's representational subsystem. The energy is depicted separately for correct and incorrect responses as well as for designated and opposing orientations. (**B**) Binned choice probabilities (clockwise) for observed data (black) and model predictions (red) as a function of the model-derived DV (gry: logistic fit to data). (**C**) Correspondence between participants' binned confidence ratings and model-based decisional certainty (grey: linear fit). (**D**) Change of signal and noise weights across training runs. All error bars denote SEM corrected for between-subject variance (*Cousineau, 2005*).

The following figure supplements are available for figure 4:

**Figure supplement 1.** Validation of the representational subsystem.

**Figure supplement 2.** Choice probabilities and the corresponding model prediction for individual participants.

**Figure supplement 3.** Confidence ratings and the corresponding model prediction for individual participants.

assessed whether the model could predict participants' trial-wise confidence reports. A correlation between the model-based decisional certainty and participants' confidence reports confirmed that confidence, too, was captured by the model (mean ± SE of $r_{Pearson} = 0.32 ± 0.02$, $t_{28} = 15.4$, $p < 0.001$; *Figure 4B*; see *Figure 4—figure supplement 3* for single-subject data).

Finally, we evaluated how perceptual learning was reflected in the update of the model's sensory filter by computing the change of signal and noise weights across runs. We expected that an increase of signal weights and a decrease of noise weights over the course of the training session was responsible for perceptual improvements. As depicted in *Figure 4C*, we found a linear increase for signal weights across runs (mean ± SEM of slope = 0.0147 ± 0.0036, $t_{28} = 4.1$, $p < 0.001$), and a linear decrease for noise weights (slope = −0.0036 ± 0.0010, $t_{28} = −3.5$, $p = 0.001$). Furthermore, the individual contrast threshold learning slopes correlated negatively with the slopes of signal

weights ($r_{Pearson}$ = −0.45, p = 0.013) and positively with the slopes of noise weights ($r_{Pearson}$ = 0.46, p = 0.011). Thus, individual learning was well captured by the signal and noise weights of the model.

## Model-free analysis of brain activation

We reasoned that if confidence-based internal feedback and reward-based external feedback share a common neural basis, neural responses in the ventral striatum to high-, average- and low-confidence events should exhibit a qualitatively similar pattern as reported for rewarding, neutral and punishing outcomes in reward-based learning (*Delgado et al., 2000*; *Knutson et al., 2001*). The results of these previous studies suggest that striatal activation reflects a positive anticipatory response at the beginning of a trial as well as a subsequent prediction error response related to the outcome. To simulate the BOLD response that arises from such a scheme, we convolved vectors coding the neural activation for an initial anticipatory response and three different outcome scenarios (positive, absent and negative prediction error) with a canonical double-gamma hemodynamic response function (*Figure 5A*). In accordance with the fMRI results of these previous studies, the simulation shows (i) an increase of striatal BOLD responses related to trial onset reflecting the anticipatory signal, and (ii) a subsequent positive, absent, or negative deflection of the BOLD response reflecting prediction errors.

To relate the neural signature of confidence in the present study to the simulation and to the results of previous reward-based studies, we binned the data into tertiles of the behavioral confidence rating (low, middle, and high confidence) and extracted the average BOLD time course in an anatomical mask of the ventral striatum using the SPM toolbox rfxplot (*Gläscher, 2009*). As shown in *Figure 5B*, the obtained event-related BOLD time courses are in remarkable agreement with the predictions of the simulation of *Figure 5A* and previous empirical findings of reward studies (*Delgado et al., 2000*). Specifically, 4–6 s after *trial onset* (reflecting the hemodynamic delay), the BOLD time courses exhibited a first peak, consistent with an anticipatory confidence signal at the start of a trial; 4–6 s after *stimulus onset*, the BOLD time courses displayed a positive deflection for high-confidence trials and a negative deflection for low-confidence trials. A statistical analysis confirmed above-baseline striatal activation at trial onset, indicative of an anticipatory signal (left peak at [−10 14 −6], $t_{28}$ = 6.42, $p_{rFWE}$ < 0.001; right peak at [12 14 −8], $t_{28}$ = 7.78, $p_{rFWE}$ < 0.001), as well as a main effect of confidence at stimulus onset in the bilateral ventral striatum (left peak at [−10 14 −4], $t_{28}$ = 10.56, $p_{rFWE}$ < 0.001; right peak at [16 12 −8], $t_{28}$ = 11.46, $p_{rFWE}$ < 0.001). This model-free assessment provides initial support for the idea that reinforcement based on reward and based on confidence share a common neural substrate both in the anticipation and the outcome period. In addition, the results lend plausibility to a model utilizing confidence as a reinforcement signal.

## Model-based analysis of brain activation

### Expected confidence

To link the confidence-based reinforcement learning model to brain activity, we estimated a new general linear model (GLM) using two time-varying parametric variables generated from the model: expected confidence at trial onset and CPE at stimulus onset. As conjectured, we found a significant parametric modulation of striatal activation by expected confidence at trial onset (right peak at [8 14 −4], $t_{28}$ = 4.12, $p_{rFWE}$ = 0.018; left peak at [−12 20 −2], $t_{28}$ = 3.97, $p_{rFWE}$ = 0.026; *Figure 5C*). This model-based result corroborates the notion that striatal activation at trial onset reflects anticipated confidence for the upcoming stimulus presentation, analogous to previous findings of an anticipatory reward signal in the ventral striatum (*Delgado et al., 2000*; *Preuschoff et al., 2006*; *Knutson et al., 2001*).

### Confidence prediction error

Using the parametric CPE regressor, we next tested for a positive linear relationship between BOLD and the CPE at the time of stimulus presentation (*Figure 5D*). As hypothesized, the bilateral ventral striatum showed a strong positive relationship with the CPE (left peak at [−16 8 −10], $t_{28}$ = 7.64, $p_{rFWE}$ < 0.001; right peak at [16 14 −8], $t_{28}$ = 7.81, $p_{rFWE}$ < 0.001). This modulation was also present in our second region of interest, the ventral tegmental area (peak at [−6 −22 −16], $t_{28}$ = 3.02, $p_{rFWE}$=0.027; see *Supplementary file 2* for a whole-brain list of active brain regions). A

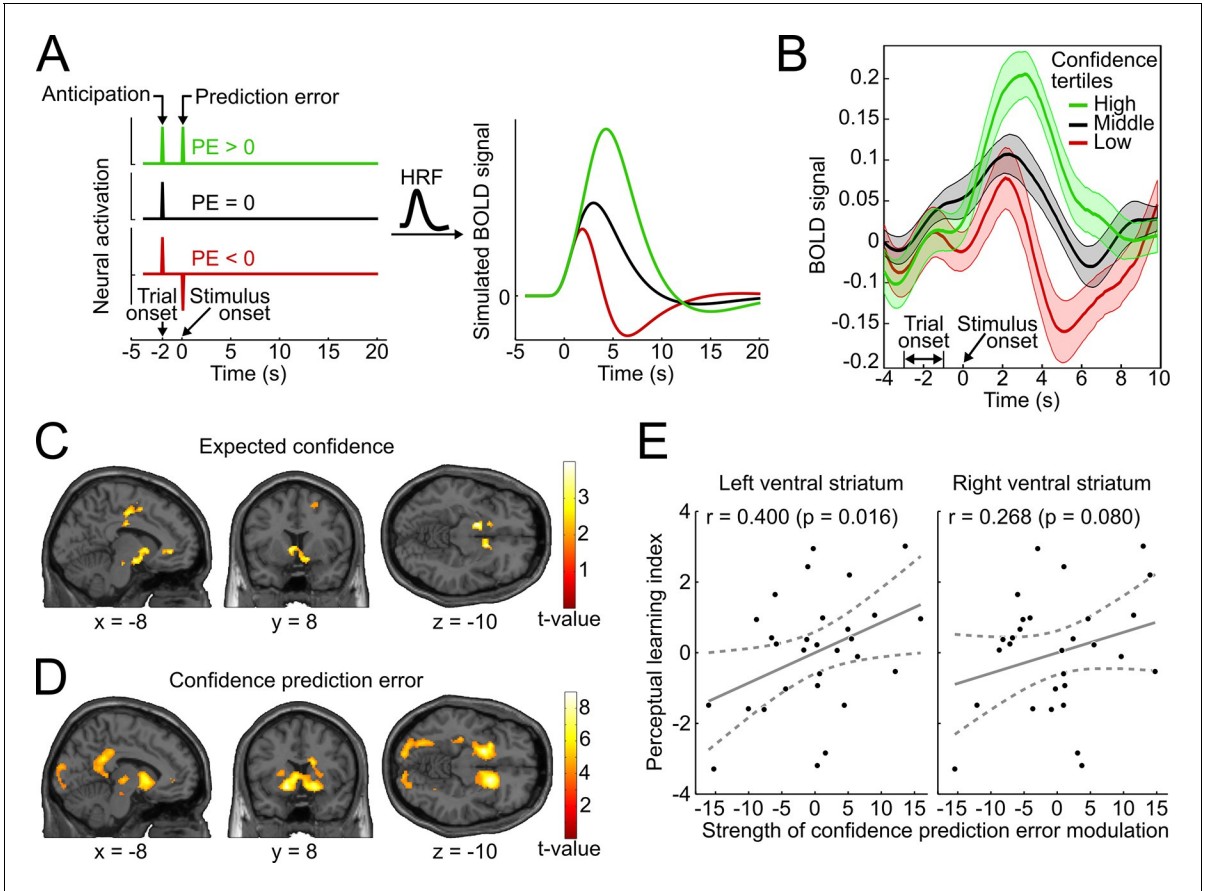

**Figure 5.** Confidence signals in the mesolimbic system and their relation to perceptual learning. (**A**) Neural activation time courses consisting of an anticipatory peak at trial onset and a positive, absent, or negative reward prediction error (PE) during outcome (stimulus onset). To simulate the associated BOLD response, the time courses were convolved with the standard canonical hemodynamic response function provided by SPM. (**B**) Event-related BOLD time courses in the ventral striatum for three tertiles of the behavioral confidence reports (representing 'low', 'middle' and 'high' confidence trials). The shaded areas denote SEM. (**C, D**) Whole-brain t-maps showing brain regions with a positive relationship between BOLD signal and expected confidence at trial onset (**C**), and between BOLD signal and CPE at stimulus onset (**D**). The t-maps were thresholded at p<0.005 (**C**) and p<0.001 (**D**), uncorrected, for illustration purposes. (**E**) Scatter plot for the relation between the strength of striatal modulation by confidence prediction errors (peak values, after age correction) and individual perceptual learning success.

The following figure supplement is available for figure 5:

**Figure supplement 1.** Control analyses accounting for effects of absolute orientation energy.

complementary analysis ruled out that this modulation was due to stimulus salience (*Figure 5—figure supplement 1*). In sum, the CPE was represented by the same key brain structures that have been implicated in signaling reward prediction errors (*Schultz et al., 1997*; *O'Doherty et al., 2004*; *Berns et al., 2001*).

## Orientation energy and decision value

To assess whether the model variables associated with the sensory and decisional subsystem would be reflected in brain activity, we performed a multivariate whole-brain searchlight (*Kriegeskorte et al., 2006*) analysis using cross-validated MANOVA (*Allefeld and Haynes, 2014*). As a variable describing the sensory evidence, we used *orientation energy* (OE), defined as the sign of the stimulus orientation (ccw = −1, cw = 1) in conjunction with the corresponding orientation energy tertile (low = 0.5, middle = 1.5, high = 2.5). We then searched for brain areas showing a multivariate linear relationship with OE. We found that OE was encoded in left occipital cortex (peak at [−44 70 0], $t_{28}$ = 4.77, $p_{cFWE}$ = 0.033), an area overlapping with voxels activated by the stimulus

localizer (*Figure 6A*). An additional ROI analysis to track the distinctness of stimulus-coding patterns across training runs within the control condition (constant contrast) was not feasible due to a lack of statistical power, i.e., decoding of orientation energy within a localizer-based visual cortex ROI was not possible when the data were restricted to the constant contrast condition (mean pattern distinctness ± SEM, D = 0.0029 ± 0.0033, $t_{28}$ = 0.86, p = 0.198, one-tailed t-test against the null hypothesis of a pattern distinctness equal to zero).

For the analysis of the decision value (DV), trials were sorted in an analogous manner, such that negative and positive DVs (reflecting the model's tendency for counterclockwise and clockwise choices) were separately sorted into DV tertiles. Interestingly, decoding of DV showed only a trend in sensory occipital cortices (peak at [−40 −72 26], $t_{28}$ = 3.35, p = 0.001, uncorrected). Instead, we found significant encoding of the DV in right middle frontal gyrus (peak at [32 14 44], $t_{28}$ = 5.87, $p_{FWE}$ = 0.026; *Figure 6B*), in line with a recent report (*Hebart et al., 2014*). Thus, perceptual and decision variables of our model can be mapped to visual and frontal cortex, respectively.

### Relation to individual perceptual learning success

Finally, we investigated whether the strength of the striatal confidence prediction error modulation translated into improvements in perceptual performance. For that purpose, we correlated the parameter estimates at the peaks of the bilateral striatal CPE contrast (coordinates [−16 8 −10] and [16 14 −8], see above) with an index for the perceptual learning success that quantified the threshold change for the trained reference axis while accounting for baseline thresholds. Age was included as an additional factor in the regression model to preclude confounding effects of age-related variation in striatal BOLD signal (*Duijvenvoorde et al., 2014*). As hypothesized, we found a significant relationship between the striatal CPE signal and individual perceptual learning success in the left ventral striatum ($r_{Pearson}$ = 0.400, p = 0.016, one-tailed; without age correction: $r_{Pearson}$ = 0.37, p = 0.026) and a trend in the right ventral striatum ($r_{Pearson}$ = 0.268, p = 0.080; without age correction: $r_{Pearson}$ = 0.24, p = 0.11) (see *Figure 5E*). This result is congruent with a viable role of CPE-based feedback signals for perceptual learning in the absence of external feedback.

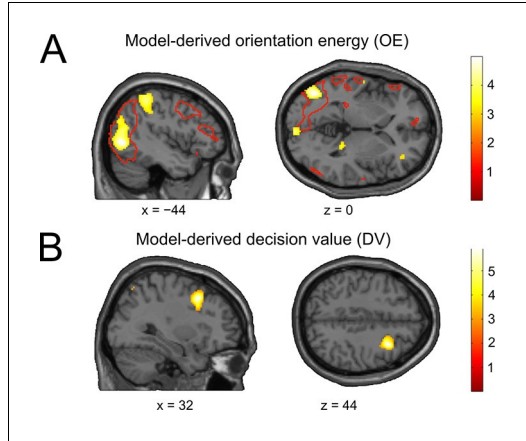

**Figure 6.** The neural basis of perceptual and decisional model variables. (**A**) Model-derived signed orientation energy (OE). The panel shows the t-map for multivariate decoding of OE. Red outlines indicate areas generally responding to the stimulus as measured with the independent stimulus localizer (t-contrast: stimulus > baseline). (**B**) Model-derived decision value (DV). T-map for multivariate decoding of the model-derived DV. All t-maps are thresholded at p < 0.005, for illustration.

## Discussion

In this study, we used perceptual learning to address the question of how humans can improve performance in the absence of external feedback. Previous reinforcement learning accounts of perceptual learning were based on external cognitive and rewarding feedback (*Law and Gold, 2009*; *Kahnt et al., 2011*) and could not explain the established phenomenon of perceptual learning without such feedback (*Herzog and Fahle, 1997*; *Gibson and Gibson, 1955*; *McKee and Westheimer, 1978*; *Karni and Sagi, 1991*). Here, we suggest that observers are capable of generating internal feedback by utilizing *confidence signals* that provide a graded evaluation of the correctness of a perceptual decision. In this way, confidence may serve as a reinforcement signal similar to reward and guide perceptual learning in cases where no external feedback is provided.

In support of this view, our model-free fMRI analyses revealed that mesolimbic confidence signals mirror those typically found for reward, both in the anticipation period (*Preuschoff et al., 2006*; *Delgado et al., 2000*; *Knutson et al., 2001*) and for prediction errors (*Schultz et al., 1997*; *O'Doherty et al., 2004*;

*Berns et al., 2001*). To establish a mechanistic ground for this suggested parallel, we devised an associate reinforcement learning model, which links behavior to computational variables that each account for a different aspect of the learning process. CPEs served as feedback in the model, defined as the difference between the current level of confidence and a long-term estimate of expected confidence. The model successfully described the learning process as a continuous adjustment of a perceptual filter linking sensory and decision units. Our model-based fMRI analyses confirmed and extended the results of the model-free analyses by demonstrating a parametric modulation in mesolimbic brain areas both by expected confidence and confidence prediction. Importantly, the strength of the striatal modulation by CPEs predicted participants' perceptual improvements, further corroborating the behavioral relevance of these internally-generated feedback signals.

The observed pattern of confidence-related activity in the mesolimbic system, including the co-modulation of the ventral tegmental area, fit well with the prediction error hypothesis of dopamine, which posits that dopaminergic midbrain neurons and their targets respond at two time points during a learning trial (*Schultz et al., 1997*). In this framework, the first response is triggered by an outcome-predictive cue and reflects an anticipatory signal. In the case of classical reinforcement learning, such a cue may be probabilistically coupled with rewards of possibly varying magnitudes. The anticipated value of the cue is then assumed to be computed as the average reward magnitude—contingent on the cue—in previous trials (*Schultz, 2006*). Here, we argue that the same principle could hold for confidence: participants learn to anticipate a certain level of confidence for the upcoming trial based on past confidence experiences, and this anticipatory state is activated when the beginning of a new trial is indicated (equivalent to a cue). In congruence with this postulation, we indeed found a modulation of striatal activity by expected confidence at trial onsets—as previously reported for expected reward (*Preuschoff et al., 2006*; *Delgado et al., 2000*; *Knutson et al., 2001*). The second response is triggered by the actual outcome and corresponds to a prediction error signal. In classical reinforcement learning, the reward prediction error represents the difference between expected value and actual outcome. In the confidence domain, the outcome would correspond to the level of confidence calculated from the stimulus and the prediction error would be computed as the difference between expected confidence and actual confidence. Overall, our results may therefore indicate that self-generated confidence assumes the role of external reward in dopaminergic prediction-error-based reinforcement learning when no external feedback is available.

A number of previous studies have used reinforcement learning models to capture the neural underpinnings of perceptual learning (*Law and Gold, 2009*; *Kahnt et al., 2011*) and category learning (*Daniel and Pollmann, 2012*). In particular, an fMRI study by Kahnt and colleagues (*Kahnt et al., 2011*) investigated perceptual learning with external reward and found that behavioral improvements were well explained by a reinforcement learning model. Their results exhibit a notable parallel to the present findings: the authors reported stimulus information encoded in visual cortex and model-derived decision value in frontal cortices, in agreement with the findings of the present study. In addition, this previous study identified a perceptual learning-related reward prediction error in the ventral striatum, dovetailing with our finding of a perceptual learning-related confidence prediction error in the same brain region. Importantly, our combined Hebbian and reinforcement learning model extends and improves previous models in several ways. First and foremost, by implementing *confidence prediction errors* in replacement of reward prediction errors, it extends previous reward reinforcement learning models of perceptual learning (*Law and Gold, 2009*; *Kahnt et al., 2011*) to cases without feedback. Second, these previous models were based on the assumption that perceptual performance is determined by a single 'readout weight', representing the amplification of stimulus information in sensory areas. While the simplicity of these models is appealing, they are limited in the sense that negative prediction errors have an unreasonable influence on behavior: according to these models, worse-than-expected feedback reduces the readout weight, which leads to an additional reduction in performance. This property runs counter to the idea that reinforcement learning agents improve their behavior through both positive and negative prediction errors. By contrast, the *associative reinforcement learning rule* of the present model entails a behaviorally advantageous and plausible function of negative prediction errors: inhibition of sensory noise. Third, a conceptually related reinforcement learning model for perceptual categorization (*Daniel and Pollmann, 2012*) implies that stimuli exclusively activate the correct stimulus category, an assumption that disregards the fact that the ambiguity of incoming stimulus information is an essential property of perceptually

demanding tasks. In contrast, the present model utilizes a dedicated *representational subsystem* (*Petrov et al., 2005*) to estimate the activation of all implemented input units, and it is their differential activity that determines perceptual choices.

The present model and results are biologically plausible and fit well with theoretical accounts of the neural basis of learning. The associative reinforcement learning rule in the model was inspired by the three-factor learning rule (*Schultz, 2002*; *Reynolds et al., 2001*), which has been proposed to underlie the potentiation of synapses in the striatum. It proposes that changes in neural transmission in cortico-striatal synapses not only depend on coincident presynaptic and postsynaptic activity (Hebbian learning), but also on the presence of dopamine error signals. Indeed, Ashby and colleagues (*Ashby et al., 2007*; *Hélie et al., 2015*) have previously suggested that the basal ganglia, which represent the predominant site of dopaminergic synaptic plasticity, are themselves a key region for learning in perceptual tasks. They proposed that (i) the basal ganglia serve to activate the appropriate target regions in executive frontal cortices shortly after sensory cortex activation; and (ii) such basal ganglia learning is superseded by cortico-cortical Hebbian learning, once the correct cortico-cortical synapses are built. This account fits well with the present model, in which perceptual learning corresponds to the process of reweighting connections between sensory and decisional units. These considerations in combination with the present results thus lend support to the hypothesis that the optimization of perceptual read-out (as implicated by our model) could be mediated via reinforcement learning in the basal ganglia.

While our study represents a first but important step towards understanding the role of confidence signals in perceptual learning, future studies are needed to investigate in more detail the characteristics of these signals which were not addressed in the current study. First, are these signals triggered independent of whether participants have to report their level of confidence after the percept or independent of whether they receive external feedback? Investigating these questions could clarify whether the observed activity in the reward network is an automatic response or depends on the task of the observer. Second, are these learning signals independent of making a perceptual decision? In other words, are they triggered only when participants have to engage in a subsequent perceptual choice? Similarly, can these confidence signals be disentangled from choice accuracy, for instance by manipulating stimulus luminance (*Busey et al., 2000*)? An answer to this latter question would shed light on the nature of the confidence signals, i.e. whether they can also be affected by metacognitive biases.

In summary, our study devised and tested a novel model of perceptual learning in the absence of external feedback, utilizing confidence prediction errors to guide the learning process. Our analyses revealed a compelling analogy between confidence-based and reward-based feedback, suggesting a similar neural mechanism for learning with and without external feedback. Future work could investigate whether a learning mechanism based on such self-generated feedback is also applicable outside the realm of perception, where learning without feedback has likewise been a long-standing puzzle (*Köhler, 1925*).

# Materials and methods

## Participants

Thirty healthy, right-handed female participants took part in the experiment in return for payment after giving written informed consent. Participants of only one gender were selected in view of the planned between-subject analysis, because male and female brains on average have different brain volumes (*Ruigrok et al., 2014*), introducing between-subject noise in the spatial normalization procedure, and slightly different hemodynamic response profiles (*Jaušovec and Jaušovec, 2010*), which could impact the detectable BOLD signal. One participant was excluded due to fixation failure, leaving 29 valid participants (24.1 ± 2.5 years, range 19–31 years). The relatively large total sample size of 30 size was chosen in view of the planned between-subject correlation between striatal modulation and learning success. As the best available study for comparison, we determined *Schlagenhauf et al. (2013)* (N = 28), which investigated the association between striatal prediction error signaling and fluid intelligence. The sample size for the present study was estimated through a method recommended by *Hulley et al. (2013)* ($N \approx \left[ \frac{z_\alpha + z_\beta}{0.5\ln[(1+r)/(1-r)]} \right]^2 + 3$), thereby applying the

observed associative strength in this previous study (r = 0.47), a Type I error rate α = 0.05 ($Z_\alpha$ = 1.960) and a Type II error rate β = 0.2 ($Z_\beta$ = 0.842). The present study was conducted according to the declaration of Helsinki, and approved by the ethics committee of the Charité Universitätsmedizin Berlin.

## Setup and schedule

Each participant came in for four sessions. The training session took place in an fMRI scanner with a back-projection screen setup (Sanyo PLC-XT21L, 60 Hz, resolution 1024 x 768). Participants responded with their right hand using an MR-compatible trackball (Current Designs Inc., Philadelphia, PA) by pressing the left button with the thumb, the right button with the middle finger and navigating the trackball with the index finger. Around one week (7.1 ± 0.2 days) before and one day after the training session, the behavioral pre- and post-test took place in a darkened room, in which the participant sat in front of a 17'' LCD monitor (LG Flatron L1750S, 60 Hz, resolution 1024 x 768) and operated with an equivalent trackball device. Around 10 weeks (70.6 ± 2.1 days) after the training session, a long-term test was conducted with a setup identical to the pre- and post-test sessions.

## Trial

Each trial started with a fixation screen for 2000 ms ± 1000 ms, followed by the presentation of the stimulus for 100 ms. After 2000 ms ± 1000 ms the white fixation cross turned orange or blue (depending on the response mapping; see section 'Training') for 750 ms to signalize the appearance of the confidence rating scale (see subsection confidence below). After adjusting the confidence rating bar, participants made a binary judgment about the stimulus orientation using the two buttons of the trackball device. After the button press, the response screen remained up for 1 s.

## Confidence rating

To avoid a potential bias by the choice itself (*Kvam et al., 2015*; *Sniezek et al., 1990*; *Sieck, 2003*; *Tafarodi et al., 1999*), participants reported their confidence prior to reporting their choice. The confidence rating scale was visualized as a half-open circle (radius r = 4° of visual angle) that linearly increased in width (0.1° to 1° visual angle) and color (black to green), whereby the orientation and angular direction of the circle changed randomly from trial to trial. Participants received the following instruction: "After the presentation of the stimulus, a rating scale appears, on which you should indicate how confident you are that your perceived orientation matches the correct orientation of the stimulus. Placing the slider of the rating scale on the thin black end would mean that you have absolutely no confidence in your perceived orientation. Placing the slider at the thick green end would mean, that you are entirely confident about your perceived orientation. Try to rate all intermediate levels of confidence proportionally in between both ends of the scale" To select a confidence rating, participants adjusted a sliding white bar on the rating scale by means of a trackball device. No time pressure was imposed.

## Test sessions

The aim of the pre-, post- and long-term test (henceforth test sessions) was to determine individual contrast thresholds in the orientation discrimination task. The test sessions were divided into blocks of 16 trials with alternating reference axes and continued until the termination criterion of a staircase procedure was reached. Each block began with a start screen that indicated the reference axis of the upcoming block. The staircase procedure started with a one-up-one-down staircase with a relative stepsize of 0.05 log units to rapidly approximate the rough threshold range (start contrast: $c_p$ = 20%; cf. *Eq. 1*). After three reversals, the algorithm switched to a weighted one-up-two-down staircase (*Kaernbach, 1991*) for fine-tuning. The ratio of *stepsize down / stepsize up* was set to 0.5488 (*García-Pérez, 1998*) with *stepsize down* set to 0.33%, leading to convergence at a performance of 80.35 percent correct. The termination criterion was the 9th reversal. Thresholds for the horizontal and vertical references axes were independently adjusted. In order to familiarize with the stimulus materials, all participants performed an additional 8 blocks at maximal contrast ($c_p$ = 100%) prior to the pre-test.

## Training

The training session in the fMRI scanner comprised an initial adjustment run and nine training runs, each with 48 trials. During the adjustment run and all training runs, the participants viewed only one reference axis ('trained reference axis'), which was assigned to participants based on the parity of their consecutively numbered participant IDs. Participants performed the adjustment run in the scanner prior to the first training run in order to accommodate with the scanner environment and to fine-tune the initial contrast level for the training runs. The adjustment run started at the determined contrast threshold of the pre-test and was subsequently adapted with the same weighted one-up-one-down staircase procedure used in the test sessions, targeting a performance of 80.35 percent correct. In the critical condition during the training runs, performance was kept constant at 80.35 percent correct by continuously adapting stimulus contrast through the above-described one-up-one-down staircase procedure. The training runs included an additional control condition with constant stimulus contrast in an interleaved half of the trials to permit an assessment of orientation information encoded in activation patterns of visual cortex without the confound of a changing stimulus contrast. The instruction for the response mapping between stimulus orientation (counterclockwise/clockwise) and response button (left/right) was alternated between runs. The response mapping was indicated at the beginning of a run and additionally in each trial through the color (blue/orange) of the fixation cross (the assignment of color and orientation being counterbalanced across participants).

## Stimuli

The stimuli were based on an additive mixture of a Gabor patch at four possible orientations (+20° or −20° from the vertical or horizontal reference axis) and phase-randomized spectrally filtered noise (*Petrov et al., 2006*). Each stimulus consisted of a greyscale Gabor patch *G(x, y)* embedded in a larger field of filtered greyscale noise *N(x, y)* (*Petrov et al., 2006*). The luminance *L(x, y)* of each pixel was an additive mixture of a Gabor term *G(x, y)* and noise *N(x, y)*, where $L_0$ was the background luminance of the screen (51.9 Cd/m$^2$):

$$L(x,y) = \left[1 + \frac{C_p}{100}G(x,y) + \frac{C_n}{100}N(x,y)\right]L_0 \tag{1}$$

$$G(x,y) = e^{-\frac{x^2+y^2}{2\sigma^2}}\sin[2\pi f(x\cos\theta + y\sin\theta) + \varphi] \tag{2}$$

The peak target contrast $c_p$, which could take values between 0% and 100%, was continuously adapted by a staircase procedure. The Gabor patches had a fixed phase $\psi$ of 0.25, a spatial frequency *f* of 1.25 cycles per degree visual angle, and the standard deviation $\sigma$ of their Gaussian envelope was set to 0.6. The orientation $\theta$ was set to +20° or −20° from the vertical (0°) or horizontal (90°) reference axis. The Gabor patch was trimmed at values smaller than 0.005, resulting in a radius of 1.87° visual angle.

The noise field *N(x, y)* was constructed from a random phase and a bandpass-filtered power spectrum. The power spectrum was generated by subtracting two Butterworth low-pass filters (two-dimensional, resolution 300 x 300 pixel, corresponding to 5.3° x 5.3°; order 3) with cut-off frequencies one octave below and one octave above the spatial frequency of the Gabor patch. The phase spectrum was sampled as a 300 x 300 matrix of uniformly distributed random numbers. The noise contrast $c_n$ was fixed at 15%. Inverse Fourier transformation of the power and phase spectrum resulted in a noise field that effectively interfered with the spatial frequency of the Gabor patch. The additive mixture of the Gabor patch and the noise field was multiplied with a circular filter and cropped at a radius of 2.5° visual angle, such that the stimulus became circular and smoothly faded out to background luminance. The filter was constructed as the inverse of a two-dimensional 300 x 300 pixel Butterworth low-pass filter of order 7 with cut-off frequency 0.275 cycles / degree visual angle. The value of the cut-off frequency ensured that the fading zone overlapped with the outer border of the Gabor patch and the high order of the filter ensured that the fading zone was relatively steep. The luminance of the monitor in test sessions and the projection setup in the training session was equalized through pre-measured color look-up tables.

## Stimulus localizer

To independently identify stimulus-responsive regions for a multivariate analysis, we conducted a localizer run with 18 stimulus blocks and 18 baseline blocks of 12 s duration in pseudo-randomized order. In the stimulus blocks the Gabor patch was shown with maximal contrast ($c_p$ = 100%) at an eccentricity of 5° visual angle and alternated every 250 ms between phase $\psi$ and counterphase 1-$\psi$ (phase and eccentricity were identical to the test and training sessions). The orientation of the Gabor patches alternated block-wise between the two orientations shown in the training session of the respective participant (± 20° with respect to the trained reference axis). The baseline blocks consisted of the fixation cross only. To hold participants' attention, they performed an independent color change detection task on the central fixation cross. The task was to press one of the buttons of the trackball device as soon as the fixation cross turned from white to red. They were instructed that, while fixating and performing the task, they should still note and make themselves aware of the Gabor stimuli.

## Perceptual learning index

To quantify participants' perceptual learning success for the trained reference axis, the respective pre-test contrast thresholds were subtracted from post-test thresholds (threshold improvement). However, an analysis of the relationship between pre-test thresholds and threshold improvements showed a strong positive correlation ($r_{Pearson}$ = 0.73, p < 0.001), suggesting that participants starting at higher thresholds had more room for performance to improve. Thus, to correct our perceptual learning index for this substantial learning-unrelated dependency, pre-test thresholds were regressed out from threshold improvements across participants. The perceptual learning index then corresponds to the resulting residuals and has mean zero.

## Eyetracking

Eyetracking data were successfully collected in 24 participants during the fMRI training session using an infrared video eyetracking system (iView XTM MRI 50 Hz, SensoMotoric Instruments, Teltow, Germany). For six other participants, eye tracker calibration failed. As a measure of fixation reliability, we computed the percentage of recorded eye gaze positions during stimulus presentation within a circle of 2.5° visual angle in radius around the center of the fixation cross. This radius corresponded to the eccentricity of the first stimulus pixel. The cut-off for exclusion was a percentage of below 95%.

## Confidence-based perceptual learning model

### Representational subsystem

To compute the orientation energy in the stimuli in each trial, we used a Matlab toolbox developed by Petrov and colleagues (*Petrov et al., 2005*; *2006*) (online available at alexpetrov.com/proj/plearn, version 1.1.1). The relevant component of the toolbox is the "representational subsystem", which takes raw images as input and computes the stimulus energy for pre-specified orientations and spatial frequencies as output. The underlying model is based on an input layer of orientation- and frequency-selective V1 simple cells, which subsequently feed into phase- and location-invariant activation maps. The output of the model is a single energy value for each specified orientation (here: −20°, 20°, 70° or 110°) and spatial frequency (here: 1.25 cycles / degree). For the present analyses we used the location-tolerant, but unnormalized energy maps computed by the toolbox.

### Associative reinforcement learning model

The orientation energy detectors of the representational subsystem (see above) are connected to decision units through *signal weights* (connecting detectors to decision units of the *same orientation*) and *noise weights* (connecting detectors to decision units of the *opposing orientation*). The activities of clockwise ($A_{cw}$) and counterclockwise ($A_{ccw}$) decision units are computed through weighted sums over normalized (i.e., divided by the maximum energy of each participant) orientation energies ($E_{cw}$, $E_{ccw}$) of the input units:

$$A_{ccw} = E_{ccw}W_{ccw,ccw} + E_{cw}W_{cw,ccw}$$
$$A_{ccw} = E_{cw}W_{cw,cw} + E_{ccw}W_{ccw,cw}$$

(3)

The difference of these output activities constitutes the decision value *DV*:

$$DV = A_{cw} - A_{ccw}$$

(4)

In addition, the model computes its decisional certainty *c'* proportional to the absolute value of *x* with scaling parameter $\lambda$:

$$c' = \lambda|DV|$$

(5)

The decisional certainty is used to fit the model to participants' normalized confidence reports (see below), which are key for the confidence-based learning rule. The general idea of the confidence-based learning rule is to reinforce circuitry giving rise to higher-than-expected confidence and to weaken circuitry giving rise to lower-than-expected confidence. For this purpose, the model continuously estimates the expected level of confidence ($\bar{c}$) by means of a Rescorla-Wagner rule (*Eq. 6*) with learning rate $\alpha_c$. In this way, the CPE $\delta$ can be computed as the difference between actual confidence c and expected confidence $\bar{c}$ (*Eq. 7*).

$$\bar{c} \leftarrow \bar{c} + \alpha_c\delta$$

(6)

$$\delta = c - \bar{c}$$

(7)

Please note that the confidence c was normalized for each participant to the range 0.1 to make model parameters comparable across participants. The model uses an associative reinforcement learning rule (*Barto et al., 1981*) to update weights both in relation to the CPE and proportional to the correlated activity of presynaptic activations $E_{ccw/cw}$ and postsynaptic activations $A_{ccw/cw}$:

$$W_{cw,choice} \leftarrow W_{cw,choice} + \alpha_w\delta E_{cw}A_{choice}$$
$$W_{ccw,choice} \leftarrow W_{ccw,choice} + \alpha_w\delta E_{ccw}A_{choice,}$$

(8)

whereby *choice* represents either *cw* or *ccw*, i.e., indicating the observer's perceptual decision for clockwise or counterclockwise orientation, respectively. The Hebbian component ensures that the update more strongly affects those connections that contribute more to the final choice. The sign of the CPE $\delta$ determines whether connections are strengthened or weakened and its absolute value modulates the extent of the update.

## Model fit to behavior and model initialization

The likelihood for participants' choices (decisions) *d* is computed through a softmax action selection rule:

$$p(d|d=cw) = 1 - p(d|d=ccw) = \frac{1}{1 + e^{-\beta x}}$$

(9)

The likelihood for participants' confidence reports c = [0;1] is assumed to be normally distributed with standard deviation $\sigma$ around the model's decisional certainty *c'*:

$$p(c) \sim \mathcal{N}(c', \sigma) \text{ if } c \in ]0;1[$$

(10)

For the boundary cases c = 0/1 the likelihood is computed as the area under the normal density in the range]$-\infty$; 0] for c = 0, and [1; $\infty$[for c = 1, respectively.

The free model parameters ($\alpha_w, \alpha_c, \beta, \lambda, \sigma$) of each participant, as well as the initial values of the signal weights ($w_{signal}^0 = w_{ccw,ccw}^0 = w_{cw,cw}^0$) and noise weights ($w_{noise}^0 = w_{ccw,ccw}^0 = w_{cw,cw}^0$) were estimated in a two-stage maximum likelihood estimation (MLE) procedure (both stages maximized the likelihood p(*d*, c)). The first MLE stage served to estimate the initial values of the weights and was based on the pooled data of all participants. We introduced this group-level MLE stage to achieve maximal power for the estimation of the initial weight values. An initial attempt to estimate the weight values at the participant level produced unreliable estimates, likely due to the non-independence of initial noise weight values and the inverse temperature parameter $\beta$ (both parameters influence the noise in the decision process). The estimates of $w_{signal}^0$ and $w_{noise}^0$ were then used as initial values in the

second MLE stage, in which the parameters were estimated individually for each participant. See **Supplementary file 1** for results of this two-stage MLE procedure.

## FMRI data acquisition and analysis

### FMRI data acquisition

Functional MRI data were acquired on a 3-Tesla Siemens Trio (Erlangen, Germany) scanner using a gradient echo planar imaging sequence and a 12-channel head-coil. The nine experimental runs comprised an average of $201 \pm 2$ (mean $\pm$ SEM) whole-brain volumes (TR = 2 s, echo time (TE) 25 ms, flip angle 78°, 36 slices, descending acquisition, 3mm isotropic resolution, interslice gap 0.45 mm, tilt angle $-20°$ from ac–pc line). The exact number of volumes could vary from run to run and depended on participants' response times. Additionally, we recorded a high-resolution T1-weighted image (TR = 1.9 s, echo time (TE) 2.51 ms, flip angle 9°, 192 slices, resolution 1 mm isotropic) and a functional localizer run (220 volumes). Preprocessing was performed using SPM8 (www.fil.ion.ucl.ac.uk/spm) and included realignment to the first image, coregistration with the structural image, spatial normalization into the Montreal Neurological Institute reference system and smoothing with an 8 mm Gaussian kernel.

### Univariate fMRI data analysis

Two different GLMs were used to model the BOLD response for the univariate model-free (GLM1) and model-based fMRI analysis (GLM2). Both GLMs comprised onset regressors for the stimulus and the response screen and six motion regressors from the realignment analysis. The stimulus regressor was modeled as a stick function and the response screen regressor as a boxcar function with durations equal to the appearance time of the response screen. Both regressors were convolved with a canonical hemodynamic response function. For GLM1 the stimulus regressor was split into three regressors, each representing a tertile of the behavioral confidence reports in a given run (low, middle, and high confidence tertiles). In GLM2, an additional regressor for the trial onset (modeled as a stick function) was included, as well as two parametric regressors, accounting for a modulation of the trial onset regressor by expected confidence ($\overline{c}$) and a modulation of the stimulus onset regressor by the CPE ($\delta$). At the group level, GLM1 was used to test for above-baseline striatal activation at trial onset and for a main effect of confidence at stimulus onset. GLM2 was used in the model-based analysis to test for a positive linear relationship between BOLD signal and expected confidence at trial onset, and for a positive linear relationship between BOLD signal and CPEs at the time of stimulus presentation.

### Multivariate fMRI data analysis

To identify the neural basis of additional model variables, we performed a multivariate analysis of variance using the cvMANOVA toolbox (**Allefeld and Haynes, 2014**). CvMANOVA provides a cross-validated scheme to estimate the distinctness of multivoxel activation patterns and allows analyzing arbitrary estimable contrasts between experimental conditions. To this aim, additional GLMs were estimated for the analysis of orientation energy (OE) and decision value (DV). As for the univariate analyses, both GLMs included regressors for stimulus onset and the response screen, as well as six motion regressors. In case of the OE model, experimental trials across all runs were sorted into three energy tertiles for clockwise (cw) and counterclockwise (ccw) orientation, leading to the following six stimulus onset regressors (orientation/energy): ccw/high, ccw/middle, ccw/low, cw/low, cw/middle, cw/high. For the DV model we used an identical binning procedure, whereby the sign of the DV determined the orientation. As contrast matrices for the cvMANOVA we used [$-2.5\ -1.5\ -0.5\ 0.5\ 1.5\ 2.5$] for both OE and DV, i.e., we tested for brain regions exhibiting a linear multivariate relationship with OE or DV. When cvMANOVA was performed as a searchlight analysis (**Kriegeskorte et al., 2006**), spheres with a radius of four voxels (257 voxels) were used.

### Group-level inference

In all cases, the resulting first-level images (contrast images or distinctness images) were submitted to a group-level t-test. The statistical cutoff was set to $p < 0.05$, family-wise-error-corrected for multiple comparisons within a region of interest ($p_{rFWE}$), at the cluster level ($p_{cFWE}$) with a cluster-defining threshold of $p < 0.001$, or at the whole-brain level ($p_{FWE}$).

## Simulation of the BOLD time course

To simulate the BOLD time course that arises from an initial anticipatory neural response and subsequent prediction error responses, we first defined vectors coding the time course of neural activation for such scenarios. These vectors were 24 s in length (240 data points, i.e., 10 Hz sampling rate) and represented the activation between 4 s before and 20 s after stimulus onset. The vectors were all zeros, except for the trial onset at t = −2 s, where the vectors were set to +1, and for the time of stimulus presentation at t = 0 s, where the vectors were set to +1 for positive prediction errors and −1 for negative prediction errors. Subsequently, these vectors were convolved with a canonical double-gamma hemodynamic response function provided by SPM8.

## Region of interest procedures

The ventral striatum ROI was based on the Harvard-Oxford cortical and subcortical structural atlases (www.cma.mgh.harvard.edu/fsl_atlas.html) and the ventral tegmental area ROI was derived from the Talairach Atlas (www.talairach.org). For exploratory multivariate analysis, we additionally generated a functional ROI based on the stimulus localizer. In a first step, we computed a group-level functional localizer t-map based on normalized first-level contrast images (*stimulus>baseline*). After thresholding at p < 0.001, we constricted the localizer-based ROI with a combined anatomical mask of occipital cortex and fusiform gyrus (*Tzourio-Mazoyer et al., 2002*). The final ROIs were created in native subject space by intersecting the reverse-normalized localizer t-map and the anatomical mask.

## Acknowledgements

This study was supported by the Research Training Group GRK 1589/1 and grants STE 1430/6-1 and STE 1430/7-1 of the German Research Foundation (DFG). We thank S Karst for her assistance during the experiments.

# Additional information

### Funding

| Funder | Grant reference number | Author |
| --- | --- | --- |
| Deutsche Forschungsgemeinschaft | GRK 1589/1 | Matthias Guggenmos Philipp Sterzer |
| Deutsche Forschungsgemeinschaft | STE 1430/6-1 | Philipp Sterzer |
| Deutsche Forschungsgemeinschaft | STE 1430/7-1 | Philipp Sterzer |

The funders had no role in study design, data collection and interpretation, or the decision to submit the work for publication.

### Author contributions

MG, Conception and design, Acquisition of data, Analysis and interpretation of data, Drafting or revising the article; GW, Acquisition of data, Analysis and interpretation of data, Drafting or revising the article; MNH, PS, Conception and design, Analysis and interpretation of data, Drafting or revising the article

### Author ORCIDs

Matthias Guggenmos, http://orcid.org/0000-0002-0139-4123

### Ethics

Human subjects: Informed consent, and consent to publish was obtained from all subjects. The study was conducted according to the declaration of Helsinki, and approved by the ethics committee of the Charité Universitätsmedizin Berlin.

## Additional files

**Supplementary files**

• Supplementary file 1. Model parameters.

• Supplementary file 2. List of active brain regions in the model-based fMRI analysis of confidence prediction errors (CPEs).

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
