## [Decision Letter]

Thank you for submitting your work entitled "Mesolimbic confidence signals guide perceptual learning in the absence of external feedback" for consideration by *eLife*. Your article has been favorably evaluated by Jody Culham (Senior editor) and three reviewers, one of whom is a member of our Board of Reviewing Editors.

The reviewers have discussed the reviews with one another and the Reviewing Editor has drafted this decision to help you prepare a revised submission.

This study investigates how individuals learn without external feedback, such as might happen in perceptual tasks with many repeated decisions. Human subjects were trained on a perceptual-discrimination task that required them to first rate their confidence and then identify whether a Gabor patch was oriented clockwise or counter-clockwise. On average, subjects showed sustained improvements in discrimination ability that were specific to the trained orientation. They modeled behavior in terms of a noisy decision variable. In the model, confidence is scaled from the magnitude of the decision variable and perturbed with Gaussian noise. The difference between the certainty in the choice on that trial and the average certainty across all past trials is used to update the relationship between the orientation energy detectors and the decision variable via an associative learning rule. They found confidence-related signals in the mesolimbic system during training. They conclude that confidence acts in a similar way as an external reward in the context of perceptual learning.

In general, the reviewers agreed that the work addresses an interesting and timely topic and presents interesting findings. However, they also had several major concerns:

1) The reviewers had many questions about the nature of the confidence signals. Most studies that measure confidence in a decision use a post-choice design where participants first make a decision and then rate their confidence in that decision. This study employed a different design where participants first report their confidence (presumably in their percept) and then made a decision about the orientation. What was the rationale for using this different procedure? What exactly were the instructions to the subjects in terms of reporting their confidence? Moreover, did RT depend systematically on confidence and/or choice accuracy? How much variability was there in the range of confidence values used by individual subjects? Did the particular range used by a given subject affect the confidence-related BOLD signals? Was there a relationship between the range of confidence values and learning ability (e.g., did apparently more confident subjects learn faster/better)? Did the variability in confidence ratings change over trials? The model would seem to predict that variance in confidence should reduce with learning. It would also be interesting if that variability was associated with a reduction in variability in stratal activation.

2) The reviewers also raised several concerns about the interpretation of the confidence-prediction error (CPE) signals found in the brain that should be clarified. The Discussion states that the results "fit well with the prediction error hypothesis of dopamine," based on responses at two time-points: "(i) an anticipatory signal triggered by an outcome-predicting cue, and (ii) a surprise signal (prediction error) triggered by the actual outcome." However, the results presented here were for what seems like two quite different time points. The first was the encoding of anticipated confidence at the beginning of a trial, but this time point was before any cue was presented that could be used to predict a particular outcome of that trial. The second was the CPE, but this occurred at the time of the stimulus presentation, not the time of "the actual outcome." In this regard, the CPE seems more closely related to either the reward-predicting signal, or perhaps a kind of sensory oddball that would be expected to evoke a different pattern of activation, such as in right inferior prefrontal areas (e.g., Strange et al., 2000). Why were such patterns not measured here? What is the evidence that the stimulus can be thought of as a form of external reward?

3) What were participants told? Were they aware that they were in a procedure where the goal was to find a level of contrast that resulted in a constant level of performance? This learning model seems sensible for this kind of situation. But how would this model work in a method of constant stimuli where each trial would present a different stimulus of different level of contrast and thus different level of mean confidence?

4) How well did the model capture behavior for individual subjects? Did the extent to which the model fit behavior also predict the involvement of striatum?

5) It would be useful to at least discuss possible implications and/or limitations of the study design in terms of interpreting the nature of the confidence signals in the brain. For example, were any measurements taken in the scanner in which the subjects reported only the choice and not the confidence judgment? Such measurements could help characterize the extent to which the confidence-related BOLD signals required the act of reporting the confidence judgment, as opposed to a more inherent, internal representation. Likewise, it might be interesting to consider the effects of manipulating confidence independent of choice accuracy (perhaps via cues that impact confidence rather than choice accuracy such as changing luminance at test see Busey et al., 2000). This should allow one to predict changes in the anticipation and prediction-error signals, independent of choice effects.

---

## [Author Response]

*In general, the reviewers agreed that the work addresses an interesting and timely topic and presents interesting findings. However, they also had several major concerns:*

We have now revised major parts of the manuscript to address these comments.

Major changes are as follows:

We included behavioral results and model fits for individual subjects to provide a more comprehensive description at the single-subject level;We elaborated on the characteristics of the confidence reports by including a separate section for the confidence report in the Methods, by performing additional analyses suggested by the reviewers and by describing the motivation behind our specific task design;We extended and clarified our discussion of the parallel between confidence and reward signals.

*1) The reviewers had many questions about the nature of the confidence signals. Most studies that measure confidence in a decision use a post-choice design where participants first make a decision and then rate their confidence in that decision. This study employed a different design where participants first report their confidence (presumably in their percept) and then made a decision about the orientation. What was the rationale for using this different procedure? What exactly were the instructions to the subjects in terms of reporting their confidence?*

We agree that the order of the confidence report and the perceptual report is an important point and was indeed an aspect of particular consideration in the design of our experiment. We apologize for not providing more details about the reasoning behind this choice in the manuscript.

Since the main variable of interest in our experiment was confidence, our goal was to get as unadulterated and unbiased a confidence report as possible. A number of studies have shown that the sole act (or even the format) of a choice can alter participants’ reported confidence (Fischhoff et al., 1977; Ronis and Yates, 1987; Sniezek et al., 1990; Tafarodi et al., 1999; Sieck and Yates, 2001; Sieck, 2003; Kvam et al., 2015). An often reported alteration is a confirmation bias, i.e. a tendency to overestimate the confidence in a choice after engaging in that particular choice. Another issue is that the memory trace of the experienced level of confidence could be interfered with by a preceding choice task or simply be attenuated by the prolonged period until the confidence rating. To avoid such issues, we decided to implement a response design in which participants first report their confidence and only subsequently submit the actual choice. An additional advantage of the response design is that participant could ‘submit’ their selected confidence level and their subsequent choice with a single button press, which was both time-efficient and intuitive. This motivation is now included in a new Methods subsection “Confidence rating”.

All participants received a standardized instruction about the confidence reports (translation from German):

“After the presentation of the stimulus, a rating scale appears, on which you should indicate how confident you are that your perceived orientation matches the correct orientation of the stimulus. Placing the slider of the rating scale on the thin black end would mean that you have absolutely no confidence in your perceived orientation. Placing the slider at the thick green end would mean, that you are entirely confident about your perceived orientation. Try to rate all intermediate levels of confidence proportionally in between both ends of the scale.”

This detailed version of the instructions is now included in the Methods subsection “Confidence rating”.

The particular wording of “that your perceived orientation matches the correct orientation of the stimulus” was chosen, because (1) participants received a familiarization session with high-contrast versions of the stimuli (see Methods, subsection “Test sessions”), in reference to which they could judge percepts in the actual experiment, and (2) with the underlying idea in mind that it is prior knowledge of the world (in this case knowledge about the appearance of a clear version of the stimulus), which enables self-generated feedback in the absence of external feedback (see Introduction, second paragraph).

*Moreover, did RT depend systematically on confidence and/or choice accuracy?*

There was a modest relation between reaction time and confidence (mean ± SE of individual z-transformed correlation coefficients: r_pearson_ = −0.06 ± 0.02; one-sample t-test against Fisher z’ = 0: t_28_ = −3.3, p = 0.002), such that participants responded faster in trials with higher confidence. Of note, 9 out of the 29 participants showed an inverse relationship, responding faster in trials with lower confidence. The correlation with choice accuracy was not significant (r_Pearson_ = −0.02 ± 0.01, t_28_ = −1.5, p = 0.14). The reason for this minor role of reaction time is likely the fact that responses were explicitly instructed as non-speeded; in fact, there was no relevant time limit for participants (a technical time-out of 30 seconds was never reached).

The relationship between reaction time and confidence/choice is now included alongside Figure 2—figure supplement 2.

*How much variability was there in the range of confidence values used by individual subjects? Did the particular range used by a given subject affect the confidence-related BOLD signals? Was there a relationship between the range of confidence values and learning ability (e.g., did apparently more confident subjects learn faster/better)? Did the variability in confidence ratings change over trials? The model would seem to predict that variance in confidence should reduce with learning. It would also be interesting if that variability was associated with a reduction in variability in stratal activation.*

To disclose the variability in the range of confidence values, we now provide confidence distributions for each participant (Figure 2—figure supplement 2).

Please note that for modeling we did not use absolute, but normalized confidence ratings (normalized to the range 0..1) in order to make the model parameters comparable across participants (see subsection “Associative reinforcement learning model”). The crucial aspect was therefore the relative precision of confidence ratings, rather than the particular ranges per se. For the same reason, our highest priority was that participants found their own intuitive way to make accurate relative judgements on the confidence scale.

For the experiment participants were instructed to choose the lowest confidence rating for trials with absolutely no confidence and the highest confidence rating for trials with perfect confidence. We found that, allowing for a margin of 3% at the borders (the width of the slider is around 3% of the total scale), all participants chose the lowest and 19/29 subjects chose the highest confidence rating at least once. We suspect that the difference between the ends of the scale is due to participant-specific criterions for “perfect” or “100%” confidence, whereas the criterion for “no confidence” is more universal. We now performed several analyses to explore the relationship between confidence ratings and our outcome measures. In each case we tested both for a linear (Pearson) and a monotonic (Spearman) relationship.

With respect to the striatal BOLD modulation, we did not find a relationship between the individual range used by a subject (defined as max(confidence) − min(confidence)) and the confidence-related BOLD signal in the left (r_Pearson_= −0.006, p = 0.98; r_Spearman_= −0.015, p = 0.44) or right (r_Pearson_= −0.009, p = 0.96; r_Spearman_ = −0.22, p = 0.24) ventral striatum. Similarly, there was no relation between the standard deviation of confidence ratings (mean ± SEM: 0.26 ± 0.016) and the BOLD signal in the left (r_Pearson_ = −0.19, p = 0.33; r_Spearman_ = −0.18, p = 0.34) and the right (r_Pearson_ = −0.23, p = 0.24; r_Spearman_ = −0.21, p = 0.28) ventral striatum.

With respect to perceptual learning, there was no relationship between learning success and the mean confidence rating (r_Pearson_ = 0.07, p = 0.70; r_Spearman_ = −0.03, p = 0.89), the range of confidence ratings (r_Pearson_ = −0.001, p = 0.99; r_Spearman_ = 0.09, p = 0.63), or the standard deviation of confidence ratings (r_Pearson_ = −0.03, p = 0.87; r_Spearman_ = −0.01, p = 0.94). Please note that the model would not predict enhanced learning with continuously high confidence levels. Rather, according to the model optimal learning is achieved if confidence signals are high for “clear percepts “(corresponding to a low co-activation of units connected by noise weights and a high co-activation of units connected by signal weights) and confidence signals are low for “noisy percepts” (high co-activation of units connected by noise weights and a low co-activation of units connected by signal weights).

The variability in the confidence ratings did not change across runs (linear regression across runs for σ_confidence_: β = 0.00066 ± 0.00079; t-test: p = 0.41, t_28_ = −0.83). However, due to the underlying staircase procedure the variability in confidence ratings is mostly determined by (i) the (remaining) variability of the stimuli for a given contrast and (ii) noise inherent to the confidence ratings, both of which are not expected to change during the experiment.

*2) The reviewers also raised several concerns about the interpretation of the confidence-prediction error (CPE) signals found in the brain that should be clarified. The Discussion states that the results "fit well with the prediction error hypothesis of dopamine," based on responses at two time-points: "(i) an anticipatory signal triggered by an outcome-predicting cue, and (ii) a surprise signal (prediction error) triggered by the actual outcome." However, the results presented here were for what seems like two quite different time points. The first was the encoding of anticipated confidence at the beginning of a trial, but this time point was before any cue was presented that could be used to predict a particular outcome of that trial. The second was the CPE, but this occurred at the time of the stimulus presentation, not the time of "the actual outcome." In this regard, the CPE seems more closely related to either the reward-predicting signal, or perhaps a kind of sensory oddball that would be expected to evoke a different pattern of activation, such as in right inferior prefrontal areas (e.g., Strange et al., 2000). Why were such patterns not measured here? What is the evidence that the stimulus can be thought of as a form of external reward?*

The reviewers’ concern is that our suggested parallel between 1) expected confidence and expected value/reward and 2) CPE and reward prediction error may not hold because they relate to conceptually different events. In addition, they suggest an alternative reflecting a sensory oddball response. We realize that the discussion of this topic in the original version of the manuscript was rather short. We clarify this parallel below, as well as in the revised manuscript. To address the possibility of an oddball-like response, we have added an analysis investigating responses to stimulus energy per se, which replicated the findings of Strange et al. (2000).

For the case of expected confidence, it is helpful to first consider the variable expected value in reinforcement studies. In classical reinforcement learning tasks, observers learn to associate a given cue with certain reward probabilities and magnitudes. In addition, the same cue might also be associated with punishment probabilities and magnitudes. Together, the learned reward and punishment probabilities/magnitudes can be used to compute the overall expected value associated with that particular cue. The point is that in these classical instrumental tasks, expected value often does not refer to a particular outcome, but rather to the anticipated “average” outcome based on past experiences (Schultz, 2006). Here we argue that the same idea holds for confidence. Participants learn to anticipate a certain level of confidence for the upcoming trial based on past confidence experiences and this anticipatory state is activated when the beginning of a new trial is indicated (equivalent to a cue). In congruence with this postulation, we indeed found a modulation of striatal activity by expected confidence at trial onsets – just as previously reported for expected reward.

The outcome in the scenario of our task design corresponds to the actual level of confidence calculated from the stimulus. We assume that subjects calculate this confidence immediately from the stimulus presented on the screen. This assumption is supported by the finding that, for dynamic stimuli, neural activity becomes predictive of perceptual confidence 200-300 ms after stimulus onset (Kiani and Shadlen, 2009; Komura et al., 2013; Zizlsperger et al., 2014). Indeed, our results confirm that striatal activity at the time of stimulus onset was clearly modulated by confidence. Finally, since there was no subsequent feedback phase that could have been anticipated, we would find it difficult to envision why a participant would compute a reward expectation when the stimulus is presented.

We now updated and extended our Discussion accordingly to clarify the parallel between reward-based and confidence-based prediction signals:

“The observed pattern of confidence-related activity in the mesolimbic system, including the co-modulation of the ventral tegmental area, fit well with the prediction error hypothesis of dopamine, which posits that dopaminergic midbrain neurons and their targets respond at two time points during a learning trial (Schultz et al., 1997). […] Overall, our results therefore indicate that self-generated confidence assumes the role of external reward in dopaminergic prediction-error-based reinforcement learning when no external feedback is available.”

We also considered the striatal response to be due to an oddball response, e.g. corresponding to enhanced activity if the stimulus orientation was “surprisingly” visible. To this end, we performed two control analyses. In the first analysis, we ensured that the observed modulation of mesolimbic activity by CPEs was not just due to orientation energy. The results indeed confirmed that orientation energy accounted only for a small fraction of the variance (Figure 5—figure supplement 1, panel A). In addition, we have now added a second analysis, in which we explicitly tested for a modulation by orientation energy (Figure 5—figure supplement 1, new panel C). The reasoning behind this analysis was that orientation energy per se could be a better proxy for the “oddballness” of a stimulus presentation compared to the computationally more complex CPE. Using the orientation energy computed by the model as a parametric regressor, we find that the pattern of activations closely matches the results reported by Strange et al. (2000), with major peaks in right dorsolateral prefrontal cortex (peak at [32, 38, 18], t_28_ = 5.24, p_uncorrected_ = 0.000007) and fusiform gyrus (peak at [−32, −56, −12], t_28_ = 3.89, p_uncorrected_ = 0.0003). These results support the notion that oddball-like responses can be accounted for by stimulus energy, but that they are distinct from CPE-related responses.

Overall, the goal of our study was to provide evidence and proof-of-concept for a role of confidence-based feedback signals as “internal” reward reinforcement signals (1) by demonstrating the involvement of the reward network (ventral striatum, ventral tegmental area) in the computation of these signals, (2) by devising and testing a reinforcement learning model based on these signals, and (3) by testing for a relation between the strength of striatal confidence-based modulation and individual learning success. Although our results confirmed our hypotheses, we acknowledge that fMRI cannot provide causal evidence for a role of confidence-based feedback signals. Such evidence may only be provided by more invasive techniques such as optogenetic stimulation. Outside of the present study, we’d like to note that a recent study (Clos et al., 2015) provided complementary evidence for a similar phenomenology of reward and confidence, demonstrating a close relationship between confidence and the subjective pleasantness associated with a given trial.

*3) What were participants told? Were they aware that they were in a procedure where the goal was to find a level of contrast that resulted in a constant level of performance? This learning model seems sensible for this kind of situation. But how would this model work in a method of constant stimuli where each trial would present a different stimulus of different level of contrast and thus different level of mean confidence?*

Participants were instructed to perform the task, but did not receive any further information about the underlying staircase procedure in the experiment. In addition, we ensured that the variability of the stimulus visibility was high even for a fixed contrast to obscure changes in contrast due to the staircase procedure. Indeed, an additional debriefing after each fMRI experiment indicated that the participants were not aware of the staircase procedure.

The hypothesized scenario with sufficiently different inter-mixed contrast levels may indeed present a problem to the model, because the expected confidence, which serves as the baseline for the CPE computation, would become unreliable. An unreliable baseline, in turn, leads to the computation of erroneous CPEs and could ultimately impede perceptual learning. This can be exemplary seen in a case, in which a series of high-contrast trials is followed by a low-contrast trial. The high-contrast trials will have rather high levels of confidence and thus the expected confidence will increase. If, for the sake of the example, the subsequent low-contrast stimulus is associated with a surprisingly high confidence (surprising in relation to previous low-contrast trials), an optimal observer would like to trigger a strong learning signal. However, because the baseline is obscured by the preceding high-contrast (and high-confidence) trials in our example, the CPE (confidence − baseline) and thus the learning signal will be rather small or even negative.

Importantly, this scenario may not only be a problem for the model, but for humans as well. Seitz et al. (2006) tested perceptual learning in motion and orientation discrimination in the absence of external feedback and in an interleaved procedure with different difficulty levels. Their original reasoning was that perceptual learning could actually be enhanced by such a procedure, because easy trials can serve as templates based on which participants are able to generate internal feedback in more difficult trials. However, they found that perceptual learning in both tasks was abolished. Similar observations have been reported for “roving” paradigms with external feedback (Yu et al., 2004; Kuai et al., 2005; Otto et al., 2006; Zhang et al., 2008; Banai et al., 2010; Tartaglia et al., 2010), which likewise found that perceptual learning is impeded when different stimulus types or difficulties are intermixed. As pointed out by Herzog et al. (2012), these results could point to a reinforcement learning nature of perceptual learning, precisely because of unclear reward prediction error baselines in such heterogeneous task designs. Thus, just as the results of these roving studies can be seen as evidential for a reinforcement learning nature of perceptual learning with external feedback, similar paradigms without external feedback, along the lines suggested by the reviewers, could serve as interesting test cases for the present model.

*4) How well did the model capture behavior for individual subjects? Did the extent to which the model fit behavior also predict the involvement of striatum?*

We now provide plots of the model fit for all 29 participants in Figure 4—figure supplement 2 & 3 to show that the average relation between model predictions and actual behavioral ratings in Figure 4 also holds at the individual-subject level.

There was no linear relation between the individual model fits (log likelihood) and the strength of the striatal modulation (left ventral striatum: r_Pearson_ = 0.20, p = 0.29; right ventral striatum: r_Pearson_ = 0.16, p = 0.41). Since the nonlinear log transformation may have obscured a potential linear relationship, we additionally computed a rank order correlation, which confirmed the first analysis (left ventral striatum: r_Spearman_ = 0.25, p = 0.18; right ventral striatum: r_Spearman_ = 0.18, p = 0.36).

5) It would be useful to at least discuss possible implications and/or limitations of the study design in terms of interpreting the nature of the confidence signals in the brain. For example, were any measurements taken in the scanner in which the subjects reported only the choice and not the confidence judgment? Such measurements could help characterize the extent to which the confidence-related BOLD signals required the act of reporting the confidence judgment, as opposed to a more inherent, internal representation. Likewise, it might be interesting to consider the effects of manipulating confidence independent of choice accuracy (perhaps via cues that impact confidence rather than choice accuracy such as changing luminance at test see Busey et al., 2000). This should allow one to predict changes in the anticipation and prediction-error signals, independent of choice effects.

We thank the reviewers and the editor for these very interesting suggestions, which became part of a new paragraph about open questions regarding the nature of the confidence signals (Discussion):

“While our study represents a first but important step towards understanding the role of confidence signals in perceptual learning, future studies are needed to investigate in more detail the characteristics of these signals which were not addressed in the current study. […] An answer to this latter question would shed light on the nature of the confidence signals, i.e. whether they can also be affected by metacognitive biases.”